organic chemistry

piperazinylethanols, ethyl chlorides, pirlindole, NMR, XRD

**Author for correspondence:**
Vanya B. Kurteva
e-mail: vkurteva@orgchm.bas.bg

This article has been edited by the Royal Society of Chemistry, including the commissioning, peer review process and editorial aspects up to the point of acceptance.

# Spontaneous conversion of O-tosylates of 2-(piperazin-1-yl)ethanols into chlorides during classical tosylation procedure

Vanya B. Kurteva[1], Boris L. Shivachev[2]
and Rositsa P. Nikolova[2]

[1]Institute of Organic Chemistry with Centre of Phytochemistry, Bulgarian Academy of Sciences, Acad. G. Bonchev street, bl. 9, 1113 Sofia, Bulgaria
[2]Institute of Mineralogy and Crystallography 'Acad. Ivan Kostov', Bulgarian Academy of Sciences, Acad. G. Bonchev street, bl. 107, 1113 Sofia, Bulgaria

VBK, 0000-0001-8703-0066

A direct conversion of piperazinyl ethanols into chlorides via a classical O-tosylation protocol is observed. The acceleration of the transformation by the piperazine unit is demonstrated. It is found that the reaction goes via the corresponding O-tosylate, which converts spontaneously into chloride with different rate depending on the substrate structure. In the case of pirlindole derivative, partially aromatized chloride formation was observed upon prolongation and/or increased excess of tosyl chloride.

## 1. Introduction

The tosylation is a classical way to convert a hydroxyl function into a better leaving group, which is widely applied as a step in various synthetic protocols [1–10]. O-tosylates are usually stable at low temperature [11] and the side-products are associated mainly with their moisture sensitivity. However, several alternative products are isolated from the reaction at high or room temperature. Dehydrotosylation at 150–185°C is frequently used to introduce unsaturation in steroids [12–15]. A substituent-dependent formation of tosylates versus ethers has been reported [16]. Spontaneous transmutation of activated 1-phenylethyl tosylates into ethers has been explained by an attack of the alcohol hydroxyl function on the tosylate arylethyl carbocation. Replacement of steroidal O-tosyl group with chlorine has been achieved by heating at 80–90°C with

pyridinium chloride in pyridine, so-called pyridinium chloride method [17–19]. The transformation of 3-phenyl-1-propanol into tosylate or chloride by reaction with tosyl chloride in pyridine has been accomplished and adapted to laboratory classroom [20]. It was demonstrated that the reaction outcome is controlled by the reaction temperature and duration; tosylate was isolated after 2 h at 0°C, while chloride was the product after more than 24 h at room temperature. Recently, it has been demonstrated that benzyl chlorides can be obtained as single products by using 4-dimethylaminopyridine as catalyst and trimethylamine as a base at 15°C and that the reaction output is driven by the substituents at the alcohol aryl moiety [21].

Piperazine derivatives are of great importance as they exist as structural subunits in a wide range of pharmacologically active compounds [22–30]. The indispensability of new and more efficient pharmaceuticals provokes enormous synthetic efforts on generating libraries of target molecules possessing piperazine unit [31–37]. In particular, the relevance of compounds build of variable units ethylene bridged to piperazine is exemplified by numerous frequently prescribed synthetic drugs like the antihistamine agents cetirizine and hydroxyzine, antidepressant agents trazodone and flesinoxan, antipsychotic agents fluphenazine and perphenazine, etc. To the best of our knowledge, there is only one record [38] in the literature on the conversion of piperazinyl ethanols into chlorides via a tosylation reaction; 70% yield of a particular example by using tosyl chloride and triethylamine in dichloromethane at room temperature.

Herein, we report on the direct chlorination of 2-piperazinyl ethanols by tosyl chloride/pyridine system. A comparison with the reaction of 2-aryl ethanol is also performed.

## 2. Results and discussion

8-Methyl-2,3,3a,4,5,6-hexahydro-1H-pyrazino[3,2,1-jk]carbazole hydrochloride (**1**) is an antidepressant drug [39–46] prescribed mainly under names pirlindole and pyrazidole. In a search of novel efficient biologically active compounds, several pirlindole derivatives were designed in the group; some based on further functionalization of 2-hydroxyethyl derivative **2a**. The latter was obtained in two steps by a literature procedure [47] and was then submitted to O-tosylation with tosyl chloride in pyridine at 5°C, a classical protocol [48–51]. Surprisingly, the chloride **4a** was isolated instead of the desired tosylate (scheme 1). The reaction was repeated with four different lots of tosyl chloride, including freshly recrystallized from hexane reagent, but the reaction output was always the same.

The structure of the chloride **4a** was assigned on the basis of NMR spectra. The latter are almost identical with those of the alcohol **2a**. Only the signals for the side-chain methylene groups are partially shifted. One of the protons of $CH_2$ neighbouring to nitrogen, numbered as 2′, is shifted downfield in **4a** and both protons for $CH_2$ connected with chlorine (3′) give common signal at 3.62 ppm, while the group gives separate signals at 3.61 and 3.69 ppm in the proton spectrum of **2a** (figure 1). The most significant is the difference in the chemical shift of the carbon-3′ resonance, which shifts from 58.6 to 41.6 ppm upon replacement of oxygen with chlorine. The structure of **4a** was additionally confirmed by single crystal XRD (figure 2).[1] A comparison with the molecular structure of **2a** (figure 3) (see footnote 1) shows that the main difference is in the side-chain orientation, while pirlindole skeleton possess identical geometry (figure 4).

The reaction was further carried out at room temperature and the conditions were optimized. As seen in table 1, the yield is not significantly influenced by the temperature (entry 1 versus entry 2). The attempts to increase the yield by prolongation of the reaction were not successful and the yields were always commensurable (entries 2–4). However, a secondary product formation was detected upon prolongation. The latter was isolated and analysed by NMR spectroscopy. The spectra show disappearance of the signals for CH-3a and three methylene groups compared with **4a** and appearance of three additional signals for aromatic CH in both proton (electronic supplementary material, figure S1) and carbon spectra (electronic supplementary material, figure S2). On that basis and by analysing the specific interactions in two-dimensional experiments, the structure of the secondary product was assigned as the partially aromatized derivative **5**, and was confirmed by single crystal XRD (figure 5) (see footnote 1).

[1]Crystallographic data (with structure factors) for the structural analysis have been deposited with the Cambridge Crystallographic Data Centre, nos. CCDC- 1856018 (**2a**), 1555951 (**4a**), 1555952 (**4b**) and 1555950 (**5**). Copies of this information may be obtained free of charge from: The Director, CCDC, 12 Union Road, Cambridge CB2 1EZ, UK. Fax: +44(1223)336-033, e-mail: deposit@ccdc.cam.ac.uk, or www: www.ccdc.cam.ac.uk.

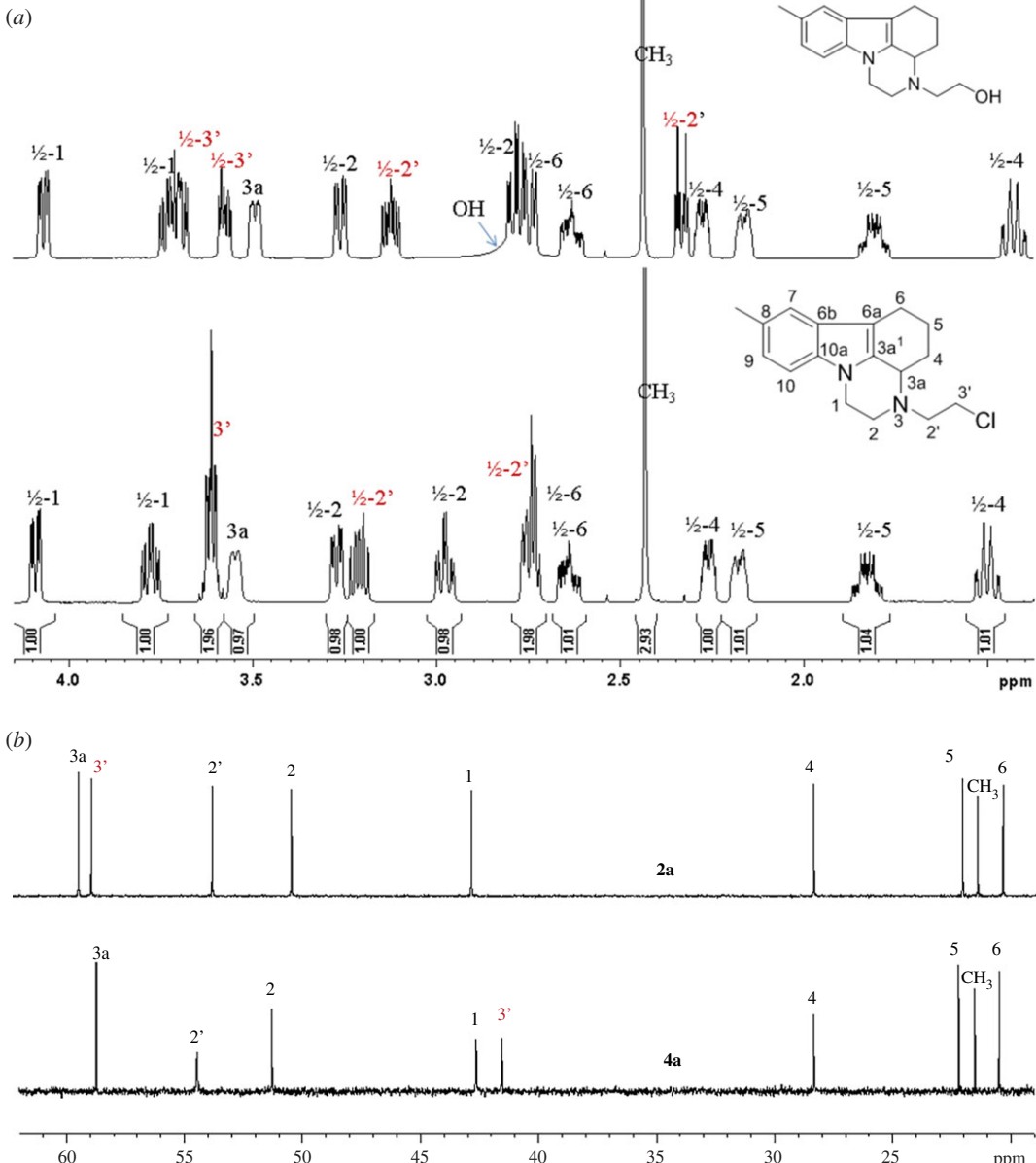

**Scheme 1.** Synthesis of chloride **4a** from alcohol **2a**.

**Figure 1.** Aliphatic areas of $^{1}$H (*a*) and $^{13}$C (*b*) NMR spectra of compounds **2a** and **4a**.

The experiment with 2.2 equivalents of tosyl chloride led to increased yield of the derivative **5** (entry 5 versus entry 4). On the other side, compound **5** was not detected by proton NMR spectra of the crude reaction mixtures before 48 h, where its content is less than 1% in respect to **4a**. So, it can be

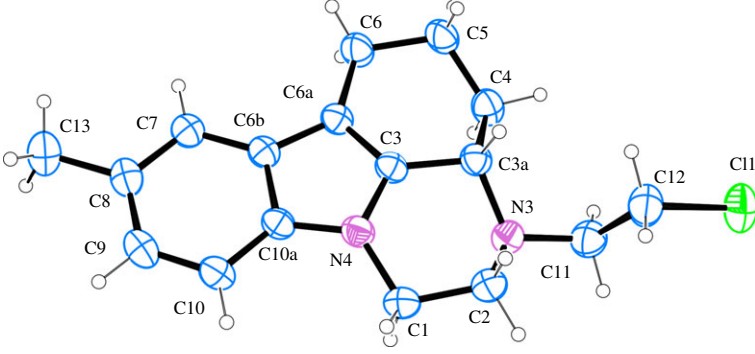

**Figure 2.** A view of the structure and the atom-numbering scheme of the independent molecule of **4a** showing 50% probability displacement ellipsoids; hydrogen atoms are shown as small spheres of arbitrary radii.

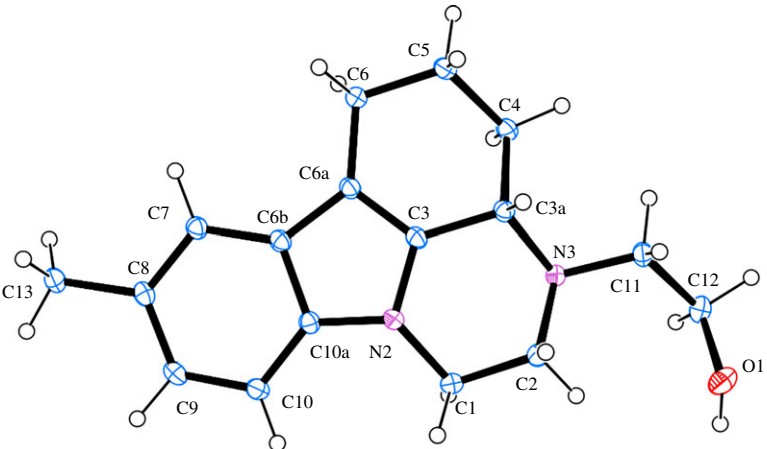

**Figure 3.** A view of the structure and the atom-numbering scheme of the independent molecule of **2a** showing 50% probability displacement ellipsoids; hydrogen atoms are shown as small spheres of arbitrary radii.

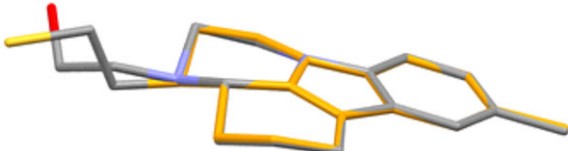

**Figure 4.** Relative orientation of the side-chain in **4a** (yellow) versus **2a** (grey).

suggested that the derivative **5** is generated by tosyl chloride or its hydrolysed derivative p-toluenesulfonic acid-assisted aromatization of **4a**. To confirm this assumption, compound **4a** was submitted to reactions with tosyl chloride and with p-toluenesulfonic acid in the same conditions. The reactions output were followed by proton NMR spectra of crude mixtures after 1, 2, 3 and 6 days. It was observed that the tosyl chloride catalysed formation of **5** is relatively fast at the beginning, 22% **5** within 24 h, and slows down afterwards; 23%, 24% and 26% **5** within 48, 72 and 144 h, respectively. On the contrary, p-toluenesulfonic acid does not catalyse the transformation in general; less than 1% of compound **5** was formed even after 6 days.

Further variation in reaction duration showed that the transformation is completed within 8 h (entry 7). No **4a** formation was detected in the spectrum of the crude product after 0.5 h (entry 11). The latter showed one main component, probably the corresponding O-tosylate **3a** ($CH_2$-3′ at 68.50 ppm; 70.89 ppm for **3f**; electronic supplementary material, figure S3), which transformed with the prolongation leading finally to the chloride **4a** (electronic supplementary material, figure S4). The assignment of the structures of the intermediate products was not reliable in the mixtures due to significant overlapping of signals. From the other side, decomposition took place during chromatography separation and pure samples were not obtained.

**Figure 5.** The structure with NMR numbering scheme and a view of the independent molecules of compound **5** showing 50% probability displacement ellipsoids; hydrogen atoms are shown as small spheres of arbitrary radii; the minor disorder component is shown as dashed bonds.

**Table 1.** Synthesis of the chloride **4a**.

| entry | conditions[a] | results |
|---|---|---|
| 1 | 5°C, 48 h | 42% **4a** |
| 2 | 48 h | 44% **4a** (0.5% **5** in respect to **4a**[b]) |
| 3 | 69 h | 46% **4a** (2% **5** in respect to **4a**[b]) |
| 4 | 87 h | 46% **4a** (7% **5** in respect to **4a**[b]) |
| 5 | 87 h, 2.2 eq. TosCl | 33% **4a**; 9% **5** (27% **5** in respect to **4a**[b]) |
| 6 | 20 h | 44% **4a** |
| 7 | 8 h | 43% **4a** |
| 8 | 6 h | **4a** and unidentified components[b] |
| 9 | 4 h | **4a** and unidentified components[b] |
| 10 | 2 h | **4a** and unidentified components[b] |
| 11 | 0.5 h | no **4a** |

[a]A solution of **2a** (10 mmol) and TosCl (11 mmol) in pyridine (15 ml) was kept at room temperature or in the refrigerator (5°C; indicated). The reaction mixture was poured into water. The solid phase formed was filtered off, washed with water, dried in desiccator, and purified by flash chromatography on silica gel.
[b]Determined by [1]H NMR spectra of the crude products.

The reaction was performed with the simplified pirlindole analogues 1-(2-hydroxyethyl) piperazine (**2b**), 1-(2-hydroxyethyl)-4-phenylpiperazine (**2c**), 1-(2-hydroxyethyl)-4-benzylpiperazine (**2d**) and 1-(2-hydroxyethyl)-4-methylpiperazine (**2e**), in the optimal conditions for **4a** in order to check its scope. As shown on table 2, the reactions with ethanols **2b** and **2c** were completed within 8 h and the corresponding chlorides **4b** and **4c** were isolated as single products in 47% and 46% yields, respectively (entries 1 and 2). In the case of alkyl-substituted piperazine ethanols **2d** and **2e**, low to negligible amount of crude products were isolated by organic solvent (entries 3 and 4) due to partial or better solubility in water-pyridine system, as detected by TLC. Benzyl-substituted chloride **4d** was

**Table 2.** Synthesis of the chlorides **4b–4g**.

| entry | starting alcohol | products | RT[a] | results |
|---|---|---|---|---|
| 1 | 2b | 4b | 8 h | 47% **4b** |
| 2 | 2c | 4c | 8 h | 46% **4c** |
| 3 | 2d | 4d | 8 h | 28% **4d** |
| 4 | 2e | 4e | 8 h | – **4e**[b] |
| 5 | 2f | X = OTos (3f), Cl (4f) | 8 h | 52% **3f**, 32% **4f**; **3f** : **4f** 1 : 0.6[c] |
| 6 | | | 40 h | 11% **3f**, 45% **4f**; **3f** : **4f** 1 : 3.8[c] |
| 7 | | | 72 h | **3f** : **4f** 1 : 24[c] |
| 8 | | | 96 h | **3f** : **4f** 1 : 83[c] |
| 9 | | | 103 h | 49% **4f** |
| 10 | 2g | 3g | 8 h | **3g**[c,d] |

[a]A solution of **2** (10 mmol) and TosCl (11 mmol; 22 mmol for **2b**) in pyridine (10 ml) was kept at room temperature. Isolation by flash chromatography.

[b]Traces of crude mixture. Not identified.

[c]Determined by $^1$H NMR spectra of the crude products.

[d]Not isolated.

isolated in less than 30% yield, while only traces of crude mixture were isolated from **2e**, for which purification and identification were not reasonable.

These results show that piperazine ethanols can be easily converted into the corresponding ethyl chlorides independent of the piperazine substitution pattern but the protocol has no practical value for water-pyridine soluble products.

Homoveratryl alcohol (**2f**) was further used in order to study the role of piperazine unit on the transformation. As seen, the reaction within 8 h led to 32% chloride **4f** and 52% of tosylate **3f** (table 2, entry 3). The reaction was prolonged and ca 1 : 4 **3f** : **4f** mixture was obtained after 40 h (entry 4), while full conversion of tosylate into chloride was achieved within 4 days (entries 5–7, electronic supplementary material, figure S5). This fact confirms the suggestion that the conversion of alcohol into chloride is going via the corresponding O-tosylate. The results also show that piperazine unit speeds up the reaction (entry 2 versus entries 5–9) most probably due to anchimeric assistance caused by nitrogen.

It has to be noted that the reaction yield is always below 50% when the conversion is complete, independent of the reaction conditions, and that the chloride **4** presents the major content of the solid/organic phase; the weight loss during the purification is less than 10%. On the contrary, when tosylate **3f** exists in the reaction mixture, the total yield of **3f** and **4f** is much higher and decreases with the consumption of tosylate; from 84% when containing 52% tosylate (entry 5) via 56% with 11% **3f** (entry 6) to 49% pure **4f** (entry 9). At the same time, the corresponding alcohol **2** was always detected in water-pyridine phase indicating that it is a reaction by-product or result of tosylate hydrolysis. However, the tosylate **3f** is stable enough to be isolated by flash chromatography on silica gel, which eliminates the assumption that the reaction yield is reduced due to tosylate hydrolysis during the water work-up. Based on these observations, it can be suggested that most probably two molecules of the initially generated O-tosylate are involved in the transformation leading to chloride **4** and alcohol **2** formation.

Finally, the behaviour of the propanol **2g** was examined in an attempt to check the dependence of the reaction output on the side-chain length. The crude mixture within 8 h contained several compounds. The main component could be assigned as tosylate **3g** based on the observed signals in the proton spectrum for methyl group and two doublets in aromatic area for two protons each (entry 10, electronic supplementary material, figure S6). Unfortunately, the product was not stable enough during chromatography purification to be explicitly characterized. This result shows that the particular protocol does not operate with piperazinyl propanols.

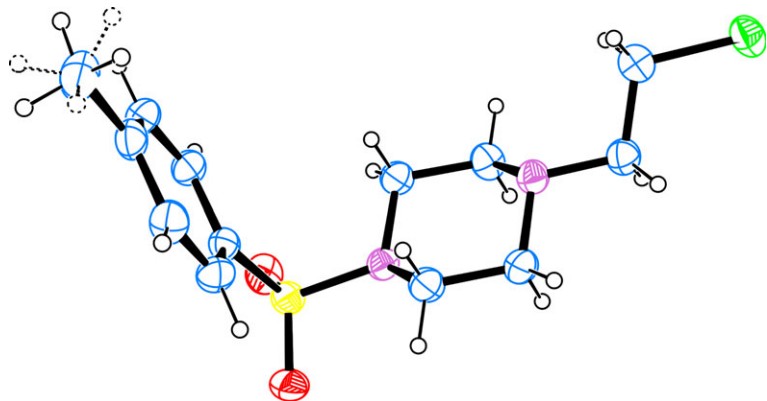

**Figure 6.** A view of the structure of the independent molecule of **4b** showing 50% probability displacement ellipsoids; hydrogen atoms are shown as small spheres of arbitrary radii; the methyl group hydrogens are disordered over two positions (represented as dashed lines).

The structures of the products were assigned by one- and two-dimensional spectra (see electronic supplementary material) and were confirmed by single crystal XRD of the chloride **4b** (figure 6) (see footnote 1).

# 3. Experimental

## 3.1. Synthesis

All reagents were purchased from Aldrich, Merck and Fluka and were used without any further purification. Fluka silica gel/TLC-cards 60778 with fluorescent indicator 254 nm were used for TLC chromatography. The melting points were determined in capillary tubes on SRS MPA100 OptiMelt (Sunnyvale, CA, USA) automated melting point system with heating rate $1°C\,min^{-1}$. The NMR spectra were recorded on a Bruker Avance II+ 600 spectrometer (Rheinstetten, Germany). The chemical shifts are quoted as δ-values in ppm using as an internal standard tetramethylsilane (TMS) and the coupling constants are reported in Hz. The assignment of the signals is confirmed by applying two-dimensional COSY, NOESY, HSQC and HMBC techniques. The spectra were processed with Topspin 3.5.6 program. The turbo spray mass spectra were taken on API 150EX (AB/MAS Sciex) mass spectrometer.

**Synthesis of starting alcohol 2a**. Pirlindole hydrochloride was partitioned between 10% aq. NaOH and DCM to obtain the free base. To a solution of **1** (0.1 mol) in dry acetonitrile (300 ml), $K_2CO_3$ (0.15 mol) and then ethyl bromoacetate (0.11 mol) were added and the mixture was stirred at room temperature (rt) for 15 h. The solid phase was filtered off and washed with acetonitrile and then with acetone. The combined organic solutions were dried over $MgSO_4$ and evaporated to dryness to give the ester derivative, which was used without purification: $R_f$ 0.42 (1% acetone/DCM); $^1$H NMR, 1.300 (t, 3H, J 7.1, $CH_3$ ester), 1.527 (qdd, 1H, J 13.7, 11.4, 2.5, $\frac{1}{2}$ $CH_2$-4), 1.836 (dddd, 1H, J 13.5, 11.5, 6.2, 2.1, $\frac{1}{2}$ $CH_2$-5), 2.168 (m, 1H, $\frac{1}{2}$ $CH_2$-5), 2.217 (m, 1H, $\frac{1}{2}$ $CH_2$-4), 2.434 (s, 3H, $CH_3$), 2.634 (dddd, 1H, J 14.4, 11.7, 6.0, 2.5, $\frac{1}{2}$ $CH_2$-6), 2.751 (ddt, 1H, J 15.6, 6.2, 1.3, $\frac{1}{2}$ $CH_2$-6), 3.208 (td, 1H, J 12.1, 4.1, $\frac{1}{2}$ $CH_2$-2), 3.320 (ddd, 1H, J 12.0, 4.8, 1.1, $\frac{1}{2}$ $CH_2$-2), 3.403 (d, 1H, J 16.7, $\frac{1}{2}$ $CH_2$-2′), 3.624 (d, 1H, J 16.7, $\frac{1}{2}$ $CH_2$-2′), 3.753 (m, 1H, $CH$-3a), 3.843 (ddd, 1H, J 12.1, 11.1, 4.8, $\frac{1}{2}$ $CH_2$-1), 4.102 (ddd, 1H, J 10.9, 4.1, 1.2, $\frac{1}{2}$ $CH_2$-1), 4.221 (qd, 2H, J 7.1, 1.2, $CH_2$ ester), 6.9790 (dd, 1H, J 8.2, 1.3, $CH$-9), 7.124 (d, 1H, J 8.2, $CH$-10), 7.238 (d, 1H, J 1.4, $CH$-7); $^{13}$C NMR 14.30 ($CH_3$ ester), 20.49 ($CH_2$-6), 21.52 ($CH_3$), 22.66 ($CH_2$-5), 27.83 ($CH_2$-4), 42.60 ($CH_2$-1), 51.25 ($CH_2$-2), 54.23 ($CH_2$-2′), 57.57 ($CH$-3a), 60.72 ($CH_2$ ester), 107.74 ($C_q$-6a), 108.78 ($CH$-10), 118.31 ($CH$-7), 122.39 ($CH$-9), 128.528 ($C_q$-6b or $C_q$-8), 128.58 ($C_q$-6b or $C_q$-8), 134.60 ($C_q$-3a$^1$ or $C_q$-10a), 134.96 ($C_q$-3a$^1$ or $C_q$-10a), 170.57 ($C$=O).

To a solution of the crude ester (0.1 mol) in THF (300 ml), $LiAlH_4$ (0.2 mol) was added and the suspension was stirred at rt for 2 h. The excess of $LiAlH_4$ was quenched with water. The solid phase was filtered off, washed with THF, dried over $MgSO_4$, and evaporated to dryness. Recrystallization from i-PrOH gave the pure alcohol **2a** as colourless solid: 78% overall yield; $R_f$ 0.27 (5% acetone/DCM); recrystallization from i-PrOH: m.p. 127.8–128.1°C (lit. [47] 121–122°C); $^1$H NMR 1.430 (qdd, 1H, J 13.6, 11.7, 2.6, $\frac{1}{2}$ $CH_2$-4), 1.810 (dddd, 1H, J 13.9, 11.7, 6.2, 2.5, $\frac{1}{2}$ $CH_2$-5), 2.163 (m, 1H, $\frac{1}{2}$ $CH_2$-5), 2.277 (m, 1H, $\frac{1}{2}$ $CH_2$-4), 2.333 (dt, 1H, J 13.0, 3.6, $\frac{1}{2}$ $CH_2$-2′), 2.438 (s, 3H, $CH_3$), 2.632 (dddd, 1H, J 14.3,

11.6, 6.0, 2.4, $\frac{1}{2}$ CH$_2$-6), 2.767 (m, 3H, $\frac{1}{2}$ CH$_2$-6 + $\frac{1}{2}$ CH$_2$-2 + OH), 3.125 (ddd, 1H, J 13.3, 9.5, 4.9, $\frac{1}{2}$ CH$_2$-2′), 3.261 (ddd, 1H, J 12.2, 4.6, 0.8, $\frac{1}{2}$ CH$_2$-2), 3.493 (ddd, 1H, J 10.6, 4.5, 2.2, CH-3a), 3.575 (ddd, 1H, J 11.1, 4.7, 3.8, $\frac{1}{2}$ CH$_2$-3′), 3.712 (m, 2H, $\frac{1}{2}$ CH$_2$-1 + $\frac{1}{2}$ CH$_2$-3′), 4.069 (ddd, 1H, J 11.0, 3.9, 0.9, $\frac{1}{2}$ CH$_2$-1), 6.979 (dd, 1H, J 8.2, 0.9, CH-9), 7.114 (d, 1H, J 8.2, CH-10), 7.250 (d, 1H, J 0.9, CH-7); $^{13}$C NMR 20.45 (CH$_2$-6), 21.52 (CH$_3$), 22.15 (CH$_2$-5), 28.37 (CH$_2$-4), 42.68 (CH$_2$-1), 50.20 (CH$_2$-2), 53.52 (CH$_2$-2′), 58.58 (CH$_2$-3′), 59.11 (CH-3a), 107.87 (C$_q$-6a), 108.79 (CH-10), 118.34 (CH-7), 122.52 (CH-9), 128.20 (C$_q$-6b), 128.72 (C$_q$-8), 134.81 (C$_q$-3a$^1$), 135.76 (C$_q$-10a).

**Synthesis of starting alcohols 2c–2e.** A suspension of 1-phenylpiperazine (10 mmol), 1-benzylpiperazine (10 mmol), or 1-methylpiperazine (10 mmol), 2-bromo ethanol (11 mmol), and K$_2$CO$_3$ (15 mmol) in dry acetonitrile (40 ml) was stirred at 90°C in closed vessel for 20 h. The solid phase was filtered off and washed with acetonitrile. The solvent was removed *in vacuo*. The product was purified by flash chromatography on silica gel by using 2% methanol in DCM as a mobile phase.

**2c:** 86% yield; $R_f$ 0.38 (10% methanol/DCM); recrystallization from heptane: m.p. 79.6–79.8°C (lit. 91°C [52], 82.5–83°C [53], 84°C [54]); $^1$H NMR 2.610 (t, 2H, J 5.4, CH$_2$-N), 2.680 (t, 4H, J 5.0, CH$_2$-2 + CH$_2$-6 piperazine), 3.211 (t, 4H, J 5.0, CH$_2$-3 + CH$_2$-5 piperazine), 3.664 (t, 2H, J 5.4, CH$_2$-OH), 6.867 (tt, 1H, J 7.3, 0.9, p-Ph), 6.935 (dd, 2H, J 8.8, 0.9, o-Ph), 7.271 (dd, 2H, J 8.7, 7.3, m-Ph); $^{13}$C NMR 49.22 (CH$_2$-3 + CH$_2$-5 piperazine), 52.88 (CH$_2$-2 + CH$_2$-6 piperazine), 57.74 (CH$_2$-OH), 59.29 (CH$_2$-N), 116.09 (o-Ph), 119.82 (p-Ph), 129.12 (m-Ph), 151.19 (i-Ph).

**2d:** 88% yield; $R_f$ 0.36 (10% methanol/DCM); colourless oil (lit. [55] colourless oil); $^1$H NMR 2.37–2.66 (bs, 8H, 4 × CH$_2$ piperazine), 2.536 (t, 2H, J 5.4, CH$_2$-N), 3.511 (s, 2H, CH$_2$-Ph), 3.601 (t, 2H, J 5.4, CH$_2$-OH), 7.251 (m, 1H, p-Ph), 7.313 (m, 4H, o-Ph + m-Ph); $^{13}$C NMR 52.85 (2 × CH$_2$ piperazine), 53.05 (2 × CH$_2$ piperazine), 57.70 (CH$_2$-OH), 59.29 (CH$_2$-N), 63.00 (CH$_2$-Ph), 127.06 (p-Ph), 128.20 (o-Ph or m-Ph), 129.19 (o-Ph or m-Ph), 137.95 (i-Ph).

**2e:** 86% yield; $R_f$ 0.19 (25% methanol/DCM); colourless oil (lit. [56] colourless oil); $^1$H NMR 2.297 (s, 3H, CH$_3$), 2.35–2.71 (bs, 8H, 4 × CH$_2$ piperazine), 2.556 (t, 2H, J 5.4, CH$_2$-N), 3.627 (t, 2H, J 5.4, CH$_2$-OH); $^{13}$C NMR 45.97 (CH$_3$), 52.80 (2 × CH$_2$ piperazine), 55.06 (2 × CH$_2$ piperazine), 57.78 (CH$_2$-OH), 59.38 (CH$_2$-N).

**Synthesis of alcohol 2 g.** A suspension of 1-phenylpiperazine (10 mmol), ethyl 3-chloro propionate (10 mmol), and K$_2$CO$_3$ (10 mmol) in dry acetonitrile (40 ml) was stirred at rt for 20 h. The solid phase was filtered off and washed with acetonitrile. The solvent was removed *in vacuo*. The product was purified by flash chromatography on silica gel by using a mobile phase with a gradient of polarity from DCM to 10% acetone in DCM: 61% ethyl 3-(4-phenylpiperazin-1-yl)propanoate [57]; colourless oil; $R_f$ 0.38 (10% acetone/DCM); $^1$H NMR 1.258 (t, 3H, J 7.1, CH$_3$ ester), 2.532 (t, 2H, J 7.4, CH$_2$-COOEt), 2.621 (t, 4H, J 5.1, CH$_2$-3 + CH$_2$-5 piperazine), 2.752 (t, 2H, J 7.4, CH$_2$-N), 3.181 (t, 4H, J 5.1, CH$_2$-2 + CH$_2$-6 piperazine), 4.146 (t, 2H, J 5.4, CH$_2$ ester), 6.846 (tt, 1H, J 7.73, 0.9, p-Ph), 6.917 (dd, 2H, J 8.7, 0.9, o-Ph), 7.253 (dd, 2H, J 8.7, 7.3, m-Ph); $^{13}$C NMR 14.23 (CH$_3$ ester), 32.32 (CH$_2$-COOEt), 49.06 (CH$_2$-2 + CH$_2$-6 piperazine), 52.91 (CH$_2$-3 + CH$_2$-5 piperazine), 53.54 (CH$_2$-N), 60.41 (CH$_2$ ester), 116.01 (o-Ph), 119.68 (p-Ph), 129.06 (m-Ph), 151.22 (i-Ph), 172.41 (C=O).

To a suspension of ethyl propanoate (5.7 mmol) in dry ether (50 ml), LiAlH$_4$ (22.8 mmol) was added portionwise and stirred at rt for 30 min. The excess of hydride was quenched by slow addition of water. The solid phase was filtered off and washed with ether. The solvent was removed *in vacuo* to give pure **2 g**: 79% yield; m.p. 75.4–75.8°C (lit. [58] 73–74°C); $R_f$ 0.18 (5% MeOH/DCM); $^1$H NMR 1.763 (m, 2H, CH$_2$-CH$_2$-CH$_2$), 2.669 (t, 2H, J 5.9, CH$_2$-N), 2.686 (t, 4H, J 4.9, CH$_2$-3 + CH$_2$-5 piperazine), 3.195 (t, 4H, J 5.1, CH$_2$-2 + CH$_2$-6 piperazine), 3.819 (t, 2H, J 5.3, CH$_2$-OH), 6.864 (tt, 1H, J 7.3, 0.8, p-Ph), 6.919 (dd, 2H, J 8.7, 0.8, o-Ph), 7.261 (dd, 2H, J 8.7, 7.3, m-Ph); $^{13}$C NMR 27.07 (CH$_2$-CH$_2$-CH$_2$), 49.21 (CH$_2$-2 + CH$_2$-6 piperazine), 53.30 (CH$_2$-3 + CH$_2$-5 piperazine), 58.74 (CH$_2$-N), 64.50 (CH$_2$-OH), 116.16 (o-Ph), 119.94 (p-Ph), 129.11 (m-Ph), 151.06 (i-Ph).

**Synthesis of chlorides 4a–4e.** *Version 1*: A solution of **2a** or **2b** (10 mmol) and TosCl (11 mmol for **2a** or 22 mmol for **2b**) in pyridine (15 ml for **2a** or 10 ml for **2b**) was kept at rt. The reaction mixture was poured into water. The solid phase formed was filtered off, washed with water and dried in desiccator. Flash chromatography purification on silica gel by using a mobile phase with a gradient of polarity from DCM to 2% acetone in DCM gave:

**4a:** colourless solid; recrystallization from i-PrOH: m.p. 124.8–125.2°C; $R_f$ 0.29 (DCM); $^1$H NMR 1.503 (qdd, 1H, J 13.6, 11.6, 2.5, $\frac{1}{2}$ CH$_2$-4), 1.830 (dddd, 1H, J 13.8, 11.7, 6.3, 2.5, $\frac{1}{2}$ CH$_2$-5), 2.178 (m, 1H, $\frac{1}{2}$ CH$_2$-5), 2.262 (m, 1H, $\frac{1}{2}$ CH$_2$-4), 2.434 (s, 3H, CH$_3$), 2.641 (dddd, 1H, J 14.2, 11.6, 6.0, 2.4, $\frac{1}{2}$ CH$_2$-6), 2.744 (m, 2H, $\frac{1}{2}$ CH$_2$-6 + $\frac{1}{2}$ CH$_2$-2′), 2.978 (td, 1H, J 12.1, 4.1, $\frac{1}{2}$ CH$_2$-2), 3.212 (ddd, 1H, J 13.8, 8.4, 7.0, $\frac{1}{2}$ CH$_2$-2′), 3.273 (ddd, 1H, J 11.9, 4.6, 1.1, $\frac{1}{2}$ CH$_2$-2), 3.545 (ddd, 1H, J 10.7, 4.6, 2.4, CH-3a), 3.613 (m, 2H, 2 × $\frac{1}{2}$ CH$_2$-3′), 3.777 (ddd, 1H, J 15.8, 11.1, 4.7, $\frac{1}{2}$ CH$_2$-1), 4.094 (ddd, 1H, J 10.9, 4.1, 1.1, $\frac{1}{2}$ CH$_2$-1), 6.970 (dd, 1H, J 8.2, 1.3,

$CH$-9), 7.116 (d, 1H, J 8.2, $CH$-10), 7.241 (d, 1H, J 1.3, $CH$-7); $^{13}C$ NMR 20.50 ($CH_2$-6), 21.54 ($CH_3$), 22.26 ($CH_2$-5), 28.35 ($CH_2$-4), 41.57 ($CH_2$-3'), 42.35 ($CH_2$-1), 51.30 ($CH_2$-2), 54.48 ($CH_2$-2'), 58.71 ($CH$-3a), 107.84 ($C_q$-6a), 108.75 ($CH$-10), 118.32 ($CH$-7), 122.44 ($CH$-9), 128.27 ($C_q$-6b), 128.64 ($C_q$-8), 134.69 ($C_q$-3a$^1$), 135.86 ($C_q$-10a); ESI (TIS)-Q $m/z$ 291 $[M + 1]^+$ (34), 289 $[M + 1]^+$ (100), 240 $[M-CH_2Cl + 1]^+$ (14), 226 $[M-CH_2CH_2Cl + 1]^+$ (9), 212 $[M-NCH_2CH_2Cl + 1]^+$ (58), 198 $[M-CH_2NCH_2CH_2Cl + 1]^+$ (28), 184 $[M-CH_2CH_2NCH_2CH_2Cl + 1]^+$ (51).

**5:** colourless solid; recrystallization from i-PrOH: m.p. 128.1–128.4°C; $R_f$ 0.77 (DCM); $^1H$ NMR 2.527 (s, 3H, $CH_3$), 3.721 (dd, 2H, J 5.1, 4.9, $CH_2$-2), 3.788 (m, 4H, $CH_2$-2' + $CH_2$-3'), 4.309 (dd, 2H, J 5.1, 4.9, $CH_2$-1), 6.605 (d, 1H, J 7.6, $CH$-4), 7.066 (t, 1H, J 7.7, $CH$-5), 7.245 (m, 2H, $CH$-9 + $CH$-10), 7.460 (d, 1H, J 7.9, $CH$-6), 7.851 (d, 1H, J 1.4, $CH$-7); $^{13}C$ NMR 21.46 ($CH_3$), 40.44 ($CH_2$-3'), 41.27 ($CH_2$-1), 48.01 ($CH_2$-2), 52.41 ($CH_2$-2'), 104.64 ($CH$-4), 108.03 ($CH$-10), 110.65 ($CH$-6), 119.67 ($CH$-5), 120.98 ($C_q$-6a), 121.05 ($CH$-7), 123.72 ($C_q$-6b), 126.52 ($CH$-9), 128.21 ($C_q$-8), 129.29 ($C_q$-3a$^1$), 132.03 ($C_q$-3a), 137.36 ($C_q$-10a); ESI (TIS)-Q $m/z$ 287 $[M + 1]^+$ (31), 285 $[M + 1]^+$ (100), 236 $[M-CH_2Cl + 1]^+$ (18), 222 $[M-CH_2CH_2Cl + 1]^+$ (6), 208 $[M-NCH_2CH_2Cl + 1]^+$ (72), 194 $[M-CH_2NCH_2CH_2Cl + 1]^+$ (43), 180 $[M-CH_2CH_2NCH_2CH_2Cl + 1]^+$ (66).

**4b:** colourless solid; recrystallization from i-PrOH: m.p. 136.1–136.3°C (lit. [59] 139–139.5°C); $R_f$ 0.34 (1% acetone/DCM), $R_f$ 0.68 (5% acetone/DCM); $^1H$ NMR 2.433 (s, 3H, $CH_3$ Tos), 2.605 (bs, 4H, $CH_2$-3 + $CH_2$-5 piperazine), 2.725 (t, 2H, J 6.6, $CH_2$-N), 3.039 (bs, 4H, $CH_2$-2 + $CH_2$-6 piperazine), 3.526 (t, 2H, J 76.6, $CH_2$-Cl), 7.326 (d, 2H, J 8.1, $CH$-3 + $CH$-5 Tos), 7.632 (d, 2H, J 8.2, $CH$-2 + $CH$-6 Tos); $^{13}C$ NMR 21.52 ($CH_3$ Tos), 40.71 ($CH_2$-Cl), 45.87 ($CH_2$-2 + $CH_2$-6 piperazine), 52.13 ($CH_2$-3 + $CH_2$-5 piperazine), 59.19 ($CH_2$-N), 127.87 ($CH$-2 + $CH$-6 Tos), 129.69 ($CH$-3 + $CH$-5 Tos), 132.16 ($C_q$-1 Tos), 143.80 ($C_q$-4 Tos).

*Version 2*: A solution of **2c**, **2d**, **2e** or **2 g** (10 mmol) and TosCl (11 mmol) in pyridine (10 ml) was kept at rt. The reaction mixture was poured into water. The products were partitioned between water and ethyl acetate. The organic phase was dried over $MgSO_4$ and evaporated to dryness. Flash chromatography purification on silica gel by using a mobile phase with a gradient of polarity from DCM to 2% acetone in DCM gave:

**4c:** colourless solid; recrystallization from heptane: m.p. 57.6–58.2°C (lit. [60] colourless oil); $R_f$ 0.26 (1% acetone/DCM), $R_f$ 0.58 (5% acetone/DCM); $^1H$ NMR 2.653 (t, 4H, J 5.0, $CH_2$-2 + $CH_2$-6 piperazine), 2.766 (t, 2H, J 7.0, $CH_2$-N), 3.188 (t, 4H, J 5.0, $CH_2$-3 + $CH_2$-5 piperazine), 3.600 (t, 2H, J 7.0, $CH_2$-Cl), 6.847 (tt, 1H, J 7.3, 0.8, p-Ph), 6.908 (dd, 2H, J 8.8, 0.9, o-Ph), 7.248 (ddt, 2H, J 8.7, 7.3, 0.7, m-Ph); $^{13}C$ NMR 40.89 ($CH_2$-Cl), 49.07 ($CH_2$-3 + $CH_2$-5 piperazine), 53.16 ($CH_2$-2 + $CH_2$-6 piperazine), 59.80 ($CH_2$-N), 116.10 (o-Ph), 119.76 (p-Ph), 129.08 (m-Ph), 151.23 (i-Ph).

**4d:** colourless oil (lit. 40–41°C [61]); $R_f$ 0.24 (5% acetone/DCM); $^1H$ NMR 2.49 (bs, 4H, $2 \times CH_2$ piperazine), 2.54 (bs, 4H, $2 \times CH_2$ piperazine), 2.722 (t, 2H, J 7.0, $CH_2$-N), 3.569 (t, 2H, J 7.0, $CH_2$-Cl); $^{13}C$ NMR 40.83 ($CH_2$-Cl), 52.72 ($2 \times CH_2$ piperazine), 52.89 ($2 \times CH_2$ piperazine), 59.70 ($CH_2$-N), 62.85 ($CH_2$-Ph), 127.23 (p-Ph), 128.26 (o-Ph or m-Ph), 129.33 (o-Ph or m-Ph), 137.38 (i-Ph).

*Version 3*: A solution of **2f** (10 mmol) and TosCl (11 mmol) in pyridine (10 ml) was kept at rt. The reaction mixture was poured into water. The products were partitioned between water and ethyl acetate. The organic phase was washed with 5% aq. HCl, and then with brine, was dried over $MgSO_4$, and evaporated to dryness. Flash chromatography purification on silica gel by using a mobile phase with a gradient of polarity from DCM to 2% acetone in DCM gave:

**4f:** colourless oil (lit. m.p. 37.5–39.5°C [62], colourless oil [63]); $R_f$ 0.45 (DCM); $^1H$ NMR 2.996 (t, 2H, J 7.5, $CH_2$-2'), 3.679 (t, 2H, J 7.5, $CH_2$-3'), 3.848 (s, 3H, $OCH_3$-4), 3.866 (s, 3H, $OCH_3$-3), 6.733 (d, 1H, J 2.0, $CH$-2), 6.754 (dd, 1H, J 8.1, 1.9, $CH$-6), 6.810 (d, 1H, J 8.1, $CH$-5); $^{13}C$ NMR 38.78 ($CH_2$-2'), 45.21 ($CH_2$-3'), 55.81 ($OCH_3$), 55.85 ($OCH_3$), 111.19 ($CH$-5), 111.98 ($CH$-2), 120.80 ($CH$-6), 130.62 ($C_q$-1), 147.89 ($C_q$-4), 148.87 ($C_q$-3).

**3f:** colourless solid; recrystallization from hexane: m.p. 46.1–46.7°C (lit. 48–50°C [64], 49–50°C [65]); $R_f$ 0.16 (DCM); $^1H$ NMR 2.419 (s, 3H, $CH_3$ Tos), 2.887 (t, 2H, J 6.9, $CH_2$-2'), 3.797 ($OCH_3$-3), 3.845 ($OCH_3$-4), 4.189 (t, 2H, J 6.9, $CH_2$-3'), 6.591 (d, 1H, J 1.9, $CH$-2), 6.654 (dd, 1H, J 8.1, 1.9, $CH$-6), 6.745 (d, 1H, J 8.1, $CH$-5), 7.265 (d, 2H, J 8.1, $CH$-3 + $CH$-5 Tos), 7.660 (d, 2H, J 8.3, $CH$-2 + $CH$-6 Tos); $^{13}C$ NMR 21.61 ($CH_3$ Tos), 34.91 ($CH_2$-2'), 55.73 ($OCH_3$), 55.88 ($OCH_3$), 70.89 ($CH_2$-3'), 111.16 ($CH$-5), 111.90 ($CH$-2), 121.00 ($CH$-6), 127.79 ($CH$-2 + $CH$-6 Tos), 128.74 ($C_q$-1), 129.74 ($CH$-3 + $CH$-5 Tos), 132.85 ($C_q$-1 Tos), 144.71 ($C_q$-4 Tos), 147.91 ($C_q$-4), 148.84 ($C_q$-3).

## 3.2. Crystallography

The crystals of **2a**, **4a**, **4b** and **5** were mounted on a glass capillary and all geometric and intensity data were taken from these crystals. Diffraction data were taken on an Agilent SuperNova Dual diffractometer

**Table 3.** Crystal data and the most important structure refinement indicators for compounds **2a**, **4a**, **5** and **4b**.

| identification code | compound **2a** | compound **4a** | compound **5** | compound **4b** |
|---|---|---|---|---|
| empirical formula | $C_{17}H_{22}N_2O$ | $C_{17}H_{21}ClN_2$ | $C_{17}H_{17}ClN_2$ | $C_{13}H_{19}ClN_2O_2S$ |
| formula weight | 270.36 | 288.81 | 284.78 | 302.81 |
| temperature (K) | 150 | 290 | 290 | 290 |
| crystal system | monoclinic | monoclinic | triclinic | monoclinic |
| space group | $P2_1/c$ | $P2_1/c$ | $P\text{-}1$ | $P2_1/c$ |
| $a$ (Å) | 10.2941(4) | 11.3939(8) | 8.7370(7) | 10.3493(3) |
| $b$ (Å) | 17.0333(5) | 13.0803(9) | 11.4112(8) | 8.5753(2) |
| $c$ (Å) | 8.0847(3) | 10.4613(5) | 15.0514(11) | 17.1092(5) |
| $\alpha$ (°) | 90 | 90 | 87.907(6) | 90 |
| $\beta$ (°) | 102.211(4) | 101.461(5) | 89.359(6) | 102.100(3) |
| $\gamma$ (°) | 90 | 90 | 80.105(6) | 90 |
| unit cell volume (Å$^3$) | 1385.51(9) | 1528.02(17) | 1477.29(19) | 1484.68(7) |
| $Z$ | 4 | 4 | 4 | 4 |
| $\rho_{calc}$ (g/cm$^3$) | 1.296 | 1.245 | 1.270 | 1.344 |
| $\mu$ (mm$^{-1}$) | 0.081 | 0.204 | 0.211 | 0.358 |
| F(000) | 584.0 | 616.0 | 600.0 | 640.0 |
| crystal size (mm$^3$) | $0.25 \times 0.22 \times 0.20$ | $0.25 \times 0.2 \times 0.15$ | $0.25 \times 0.2 \times 0.18$ | $0.35 \times 0.2 \times 0.19$ |
| radiation | MoK$\alpha$ ($\lambda = 0.71073$) | MoK$\alpha$ ($\lambda = 0.71073$) | MoK$\alpha$ ($\lambda = 0.71073$) | MoK$\alpha$ ($\lambda = 0.71073$) |
| $\Theta$ range for data collection (°) | 6.268–58.77 | 6.676–50.04 | 6.412–50.046 | 5.63–65.026 |
| index ranges | $-11 \leq h \leq 14$, $-15 \leq k \leq 23$, $-10 \leq l \leq 8$ | $-11 \leq h \leq 12$, $-15 \leq k \leq 9$, $-12 \leq l \leq 11$ | $-10 \leq h \leq 10$, $-13 \leq k \leq 11$, $-17 \leq l \leq 17$ | $-15 \leq h \leq 14$, $-12 \leq k \leq 12$, $-24 \leq l \leq 25$ |
| reflections collected/ independent | 8495/3318 | 8729/2588 | 8058/4890 | 16572/4975 |
| $R_{int}/R_{sigma}$ | 0.0252/0.0297 | 0.0375/0.0328 | 0.0300/0.0406 | 0.0234/0.0229 |
| data/restraints/ parameters | 3318/0/296 | 2588/0/187 | 4890/0/423 | 4975/0/172 |
| goodness-of-fit on $F^2$ | 1.056 | 1.015 | 1.076 | 1.082 |
| $R_1$, $wR_2$ indexes, $I \geq 2\sigma$ $(I)$ | 0.0447, 0.1101 | 0.0649, 0.1638 | 0.0683, 0.1631 | 0.0496, 0.1306 |
| $R_1$, $wR_2$ indexes, all data | 0.0527, 0.1169 | 0.0869, 0.1817 | 0.0955, 0.1819 | 0.0727, 0.1469 |
| largest diff. peak/ hole/e Å$^{-3}$ | 0.28/$-0.27$ | 0.27/$-0.25$ | 0.27/$-0.23$ | 0.35/$-0.47$ |
| CCDC number | 1856018 | 1555951 | 1555950 | 1555952 |

equipped with an Atlas CCD detector using micro-focus Mo Kα radiation ($\lambda = 0.71073$ Å) at room temperature. The determination of the unit cell parameters, data collection and reduction were performed with Crysalispro software [66]. The structures were solved by direct methods and refined by the full-matrix least-squares method on $F^2$ with ShelxS and ShelxL 2016/6 programs [67]. All non-hydrogen atoms, including solvent molecules, were located successfully from Fourier maps and were refined anisotropically. The H atoms were placed in idealized positions (C–H = 0.86 to 0.97 Å) and were constrained to ride on their parent atoms, with $U_{iso}(H) = 1.2U_{eq}(C)$. The most important crystallographic and refinement indicators are listed on table 3.

# 4. Conclusion

A direct conversion of ethanols into chlorides via a classical O-tosylation protocol is observed. It is found that:

— 2-Substituted ethanols can be easily converted into the corresponding ethyl chlorides via a simple cheap protocol.

— The reaction goes via initial *O*-tosylate formation.

— The presence of piperazine fragment at the end of ethanol unit speeds up the conversion. The transformation of *O*-tosylate into chloride is very fast in piperazinyl ethanols and slower in aromatic ethanols.

— The protocol has practical value only when the product possesses limited solubility in water-pyridine system.

— The prolongation of the reaction and/or increased excess of tosyl chloride lead to partial aromatization of pirlindole chloride. Tosyl chloride catalyses the transformation.

— The particular conditions are not applicable to piperazinyl propanols.

This study aims to warn the synthetic community about the eventual problems if trying to tosylate 2-hydroxyethyl derivatives, especially piperazinyl ethanols, and to inform on the possibility to convert directly ethanols into ethyl chlorides.

Data accessibility. All experimental and analytical data for this work are presented within the manuscript. Some additional figures and the original NMR spectra are presented in the electronic supplementary material.

Authors' contributions. The synthetic experiments and NMR analyses were carried out by V.B.K. The single crystal XRD was performed by B.L.S. and R.P.N. All authors contributed in the discussion of the results and in the manuscript writing.

Competing interests. The authors declare no competing interests.

Funding. The financial support came from The Bulgarian Science Fund, project DCOST-01-23.

Acknowledgements. The financial support by The Bulgarian Science Fund, DCOST-01-23 and infrastructure projects UNA-17/2005, DRNF-02-13/2009, and DRNF-02/01, and by The EU, COST Action CA15106 'C–H Activation in Organic Synthesis' (CHAOS), is gratefully acknowledged.

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
