## [Reviewer comments · Royal Society Open Science]

Review History

RSOS-181840.R0 (Original submission)

Review form: Reviewer 1 (Milton Kiefel)

Is the manuscript scientifically sound in its present form?

Yes

Are the interpretations and conclusions justified by the results?

Yes

Is the language acceptable?

Yes

Is it clear how to access all supporting data?

Yes

Do you have any ethical concerns with this paper?

No

Have you any concerns about statistical analyses in this paper?

No

Recommendation?

Accept with minor revision (please list in comments)

Comments to the Author(s)

This manuscript describes the unexpected chlorination of 2-hydroxyethanol derivatives during attempted tosylation. The authors provide a very detailed account of how they explored this unexpected result, both with respect to exploring the reaction conditions on the substrate 2a (data presented in Table 1) and also with alternative substrates (data presented in Table 2). The results from all of these reactions are well documented and clearly the spectroscopic data obtained, including X-ray data on two products, is supportive of the proposed chlorinated products being formed under these unusually mild conditions. One minor comment here is that there is a lack of mass spec data on these compounds - the presence of Cl would be clear in the mass spectra due to the two natural isotopes of chlorine, so perhaps such data could be included.

Whilst it is clear that the chlorination of these 2-hydroxyethanol substrates does indeed occur, it is perhaps a little less clear as to why. Firstly, the yields are always below 50%, which implies so participation from the tosylated species is required. It would perhaps be worthwhile for the authors to make some of the tosylated species, isolate and purify this compound (that would be compound 3 in the manuscript), and then expose the purified tosylate to the reaction conditions (TosCl/pyridine) and see if any chlorination results. I did not see this experiment described in the manuscript itself, and such an experiment may shed light on the mechanism. Furthermore, the authors comment that they used "four different lots of tosyl chloride", all giving the same outcome. But they authors have not discussed if they specifically purified the tosyl chloride, or if it was simply commercially obtained. It is well known that commercial tosyl chloride can contain tosic acid and HCl - such material usually has a m.p. 65-68°C (Armarego & Chai, "Purification of Laboratory Chemicals", 7th ed, p. 387). Crystallisation (from CHCl₃/pet ether) results in material with m.p. 67.5-68.5°C. Did the authors recrystallise any of their tosyl chloride - if not, I suggest they repeat the reported reaction with purified tosyl chloride.

Overall, I think this is an interesting result that will be of interest to a number of synthetic chemists, and so provided some minor additional experiments are undertaken (as suggested above), I think the work should be published.

Review form: Reviewer 2 (Carlos Sanhueza-Chavez)

Is the manuscript scientifically sound in its present form?

Yes

Are the interpretations and conclusions justified by the results?

Yes

Is the language acceptable?

Yes

Is it clear how to access all supporting data?

Yes

Do you have any ethical concerns with this paper?

No

Have you any concerns about statistical analyses in this paper?

No

Recommendation?

Accept with minor revision (please list in comments)

Comments to the Author(s)

Dear Colleagues,

This manuscript describes the preparation of 2-chloroethyl piperazines from the respective alcohols via treatment with tosyl chloride in pyridine. The article is enjoyable to read and the experimental work is written in way that it is easy to follow and reproduce. The compounds are well characterized and the physical data is well presented.

The reaction condition (TsCl/Py) employed for this synthetic study is typical for the preparation of tosylated derivatives and the generation of alkyl chlorides is a well-known outcome associated to primary alcohols and long reaction times. However, the piperazine ethanol substrate studied on this work shows a particularly higher rate of alkyl chloride formation compared to other substrates. The authors perform a study of the scope of this reaction using different piperazine ethanol substrates, different conditions and an aryl ethanol compound as a control molecule. The results point to the piperazine moiety as a key component for the observed high rate of alkyl chloride formation.

I would like to point some aspects I found in the manuscript that needs to be revised:

1. In the introduction, it is stated that there are no previous reports related to the preparation of 2-chloroethyl piperazines from the respective alcohol by this procedure (page 1, lines 34-37, 2nd column: To the best of our knowledge, there are no records in the literature on the conversion of piperazinyl ethanols into chlorides via a tosylation reaction.) However, in 2010 Sachin et al. (Bioconjugate Chem. 2010, 21, 2282–2288) employed the system tosyl chloride/triethylamine for transforming a 4-phenyl substituted piperazine ethanol into the respective 2-chloroethyl adduct (see compound 11 in the above reference). This publication must be cited in the manuscript and the introduction needs to be modified accordingly.
2. I would suggest including a discussion about a plausible mechanism for explaining the higher rate of alkyl chloride formation that is observed for piperazine ethanol derivatives. The neighboring tertiary amine from the piperazine ring can engage in the tosylate displacement (anchimeric assistance).
3. The results observed for compound 2f shows the key role of the piperazine moiety in the reaction rate. It would add to the quality of this work if the authors extend the study using a piperazine derivative with a longer alkyl chain (propyl or butyl).

Typos:

4. Page 1, Line 34: antipsyhotic (antipsychotic)
5. Page 2, Line 7: 4a instead of 4a (bold font)

Overall, I think that this is a good synthetic article that will help other researchers in the design of

synthetic routes for the preparation of 2-substituted ethyl piperazine derivatives and other similar compounds.

Hope you find my comments and suggestions appropriate and constructive,

Best Regards

Carlos A. Sanhueza Chávez
Assistant Professor – Medicinal Chemistry
College of Pharmacy and Health Sciences
St. John's University
8000 Utopia Parkway
Queens, New York, 11439

Decision letter (RSOS-181840.R0)

11-Dec-2018

Dear Professor Kurteva:

Title: Spontaneous conversion of O-tosylates of 2-(piperazin-1-yl)ethanols into chlorides during classical tosylation procedure
Manuscript ID: RSOS-181840

Thank you for submitting the above manuscript to Royal Society Open Science. On behalf of the Editors and the Royal Society of Chemistry, I am pleased to inform you that your manuscript will be accepted for publication in Royal Society Open Science subject to minor revision in accordance with the referee suggestions. Please find the reviewers' comments at the end of this email.

The reviewers and handling editors have recommended publication, but also suggest some minor revisions to your manuscript. Therefore, I invite you to respond to the comments and revise your manuscript.

Please also include the following statements alongside the other end statements. As we cannot publish your manuscript without these end statements included, if you feel that a given heading is not relevant to your paper, please nevertheless include the heading and explicitly state that it is not relevant to your work. We have included a screenshot example of the end statements for reference.

- Ethics statement

Please clarify whether you received ethical approval from a local ethics committee to carry out your study. If so please include details of this, including the name of the committee that gave consent in a Research Ethics section after your main text. Please also clarify whether you received informed consent for the participants to participate in the study and state this in your Research Ethics section.

OR

Please clarify whether you obtained the necessary licences and approvals from your institutional animal ethics committee before conducting your research. Please provide details of these licences and approvals in an Animal Ethics section after your main text.

OR

Please clarify whether you obtained the appropriate permissions and licences to conduct the fieldwork detailed in your study. Please provide details of these in your methods section.

- Data accessibility

It is a condition of publication that you make available the data and research materials supporting the results in the article. Datasets should be deposited in an appropriate publicly available repository and details of the associated accession number, link or DOI to the datasets must be included in the Data Accessibility section of the article (<http://royalsocietypublishing.org/instructions-authors#question17>). Reference(s) to datasets should also be included in the reference list of the article with DOIs (where available).

Please include a Data Availability section after your main text stating where supporting data are available from, or where they will be made available should your article be accepted for publication.

If you wish to submit your supporting data or code to Dryad (<http://datadryad.org/>), or modify your current submission to dryad, please use the following link:
<http://datadryad.org/submit?journalID=RSOS&manu=RSOS-181840>

- Competing interests

Please include a Competing Interests section after your main text declaring any financial or non-financial competing interests. If you have no competing interests please state 'I/we have no competing interests.'

- Authors' contributions

Please include an Authors' Contributions section at the end of your main text detailing the contribution of each author. All authors should have read and approved the manuscript before submission and this should be stated in the Authors' Contributions section.

The list of Authors should meet all of the following criteria; 1) substantial contributions to conception and design, or acquisition of data, or analysis and interpretation of data; 2) drafting the article or revising it critically for important intellectual content; and 3) final approval of the version to be published.

- Funding statement

Please include a funding section after your main text which lists the source of funding for each author.

Because the schedule for publication is very tight, it is a condition of publication that you submit the revised version of your manuscript before 20-Dec-2018. Please note that the revision deadline will expire at 00.00am on this date. If you do not think you will be able to meet this date please let me know immediately.

Best wishes,
Dr Laura Smith
Publishing Editor, Journals

On behalf of the Subject Editor Professor Anthony Stace and the Associate Editor Professor John Moses.

RSC Associate Editor:
Comments to the Author:
Revision requested by the referees should be addressed

RSC Subject Editor:
Comments to the Author:
(There are no comments.)

Reviewer comments to Author:
Reviewer: 1

Comments to the Author(s)

This manuscript describes the unexpected chlorination of 2-hydroxyethanol derivatives during attempted tosylation. The authors provide a very detailed account of how they explored this unexpected result, both with respect to exploring the reaction conditions on the substrate 2a (data presented in Table 1) and also with alternative substrates (data presented in Table 2). The results from all of these reactions are well documented and clearly the spectroscopic data obtained, including X-ray data on two products, is supportive of the proposed chlorinated products being formed under these unusually mild conditions. One minor comment here is that there is a lack of mass spec data on these compounds - the presence of Cl would be clear in the mass spectra due to the two natural isotopes of chlorine, so perhaps such data could be included.

Whilst it is clear that the chlorination of these 2-hydroxyethanol substrates does indeed occur, it is perhaps a little less clear as to why. Firstly, the yields are always below 50%, which implies so participation from the tosylated species is required. It would perhaps be worthwhile for the authors to make some of the tosylated species, isolate and purify this compound (that would be compound 3 in the manuscript), and then expose the purified tosylate to the reaction conditions (TosCl/pyridine) and see if any chlorination results. I did not see this experiment described in the manuscript itself, and such an experiment may shed light on the mechanism. Furthermore, the authors comment that they used "four different lots of tosyl chloride", all giving the same outcome. But they authors have not discussed if they specifically purified the tosyl chloride, or if it was simply commercially obtained. It is well known that commercial tosyl chloride can contain tosic acid and HCl - such material usually has a m.p. 65-68°C (Armarego & Chai, "Purification of Laboratory Chemicals", 7th ed, p. 387). Crystallisation (from CHCl₃/pet ether) results in material with m.p. 67.5-68.5°C. Did the authors recrystallise any of their tosyl chloride - if not, I suggest they repeat the reported reaction with purified tosyl chloride.

Overall, I think this is an interesting result that will be of interest to a number of synthetic chemists, and so provided some minor additional experiments are undertaken (as suggested above), I think the work should be published.

Reviewer: 2

Comments to the Author(s)
Dear Colleagues,

This manuscript describes the preparation of 2-chloroethyl piperazines from the respective

alcohols via treatment with tosyl chloride in pyridine. The article is enjoyable to read and the experimental work is written in way that it is easy to follow and reproduce. The compounds are well characterized and the physical data is well presented.

The reaction condition (TsCl/Py) employed for this synthetic study is typical for the preparation of tosylated derivatives and the generation of alkyl chlorides is a well-known outcome associated to primary alcohols and long reaction times. However, the piperazine ethanol substrate studied on this work shows a particularly higher rate of alkyl chloride formation compared to other substrates. The authors perform a study of the scope of this reaction using different piperazine ethanol substrates, different conditions and an aryl ethanol compound as a control molecule. The results point to the piperazine moiety as a key component for the observed high rate of alkyl chloride formation.

I would like to point some aspects I found in the manuscript that needs to be revised:

1. In the introduction, it is stated that there are no previous reports related to the preparation of 2-chloroethyl piperazines from the respective alcohol by this procedure (page 1, lines 34-37, 2nd column: To the best of our knowledge, there are no records in the literature on the conversion of piperazinyl ethanols into chlorides via a tosylation reaction.) However, in 2010 Sachin et al. (Bioconjugate Chem. 2010, 21, 2282-2288) employed the system tosyl chloride/triethylamine for transforming a 4-phenyl substituted piperazine ethanol into the respective 2-chloroethyl adduct (see compound 11 in the above reference). This publication must be cited in the manuscript and the introduction needs to be modified accordingly.
2. I would suggest including a discussion about a plausible mechanism for explaining the higher rate of alkyl chloride formation that is observed for piperazine ethanol derivatives. The neighboring tertiary amine from the piperazine ring can engage in the tosylate displacement (anchimeric assistance).
3. The results observed for compound 2f shows the key role of the piperazine moiety in the reaction rate. It would add to the quality of this work if the authors extend the study using a piperazine derivative with a longer alkyl chain (propyl or butyl).

Typos:

4. Page 1, Line 34: antipsyhotic (antipsychotic)
5. Page 2, Line 7: 4a instead of 4a (bold font)

Overall, I think that this is a good synthetic article that will help other researchers in the design of synthetic routes for the preparation of 2-substituted ethyl piperazine derivatives and other similar compounds.

Hope you find my comments and suggestions appropriate and constructive,

Best Regards

Carlos A. Sanhueza Chávez
Assistant Professor - Medicinal Chemistry
College of Pharmacy and Health Sciences
St. John's University
8000 Utopia Parkway
Queens, New York,

Author's Response to Decision Letter for (RSOS-181840.R0)

See Appendix A.

Decision letter (RSOS-181840.R1)

21-Jan-2019

Dear Professor Kurteva:

Title: Spontaneous conversion of O-tosylates of 2-(piperazin-1-yl)ethanols into chlorides during classical tosylation procedure

Manuscript ID: RSOS-181840.R1

It is a pleasure to accept your manuscript in its current form for publication in Royal Society Open Science. The chemistry content of Royal Society Open Science is published in collaboration with the Royal Society of Chemistry.

On behalf of the Subject Editor Professor Anthony Stace and the Associate Editor Professor John Moses.

RSC Associate Editor
Comments to the Author:
(There are no comments.)

Reviewer(s)' Comments to Author:

Appendix A

Response to Reviewers

Dear Prof. Chávez,

Thank you very much for the time you and the other associate editors have devoted to handle our submission. We are very grateful to you and Reviewers for the help permitting us to improve significantly the manuscript.

All corrections in the manuscript, inserted according to the Reviewers' recommendations, are highlighted in green. The end statements required are also included.

The Reviewer 1 suggested to isolate and purify tosylate and then to submit to a reaction with tosyl chloride. However, all attempts to isolate tosylate of a piperidine derivative were unsuccessful; decomposition took place. The only relatively stable tosylate was **3f** and so, we performed the reaction by taking regularly probes and analysing by proton NMR spectra (Table 2, entries 5-9), which gives information on the consumption of initially formed tosylate with the time. Concerning the purity of the reagent, we always use recrystallized tosyl chloride in order to eliminate HCl contamination. That time, we also performed experiments with non-recrystallized commercial lots, but received the same results.

Faithfully yours,

Vanya Kurteva